# The Family and Community Nurses Cultural Model in the Times of the COVID Outbreak: A Focused Ethnographic Study

**DOI:** 10.3390/ijerph20031948

**Published:** 2023-01-20

**Authors:** Michela Barisone, Erica Busca, Erika Bassi, Enrico De Luca, Emanuele Profenna, Barbara Suardi, Alberto Dal Molin

**Affiliations:** 1S.C. Centro Controllo Direzionale, Azienda Socio Sanitaria Ligure (ASL2), Piazza Pertini, 10, 17100 Savona, Italy; 2Department of Translational Medicine, University of Piemonte Orientale, Via Solaroli 17, 28100 Novara, Italy; 3Azienda Ospedaliero Universitaria Maggiore della Carità Novara, Corso Mazzini 18, 28100 Novara, Italy; 4Department of Nursing, Faculty of Health and Life Sciences, University of Exeter, St. Luke’s Campus, Exeter EX1 2LT, UK; 5Azienda Sanitaria Locale di Parma, Strada del Quartiere n. 2/A, 43125 Parma, Italy; 6S.C. Direction of Health Professions, Azienda Sanitaria di Vercelli, Ospedale Sant’Andrea, Corso Mario Abbiate 21, 13100 Vercelli, Italy

**Keywords:** family and community nurse, cultural model, COVID-19, primary care, qualitative research, ethnographic research

## Abstract

The ageing population, increasingly frail and chronically ill, and COVID-19 pandemic challenges have highlighted national health systems’ vulnerability and, more strongly/to a greater extent, the pivotal role of the family and community nurse (FCN). However, the recent introduction of FCNs in primary care settings has yet to be explored in Italy. This study aimed to identify the FCNs’ cultural model and its implementation during the COVID-19 outbreak. A focused ethnographic study was performed in a primary care community service in northern Italy. Participants were FCNs (N = 5), patients and caregivers (N = 12). Qualitative data were collected through semi-structured interviews, field notes, observation of FCNs’ activities and access to documents. Qualitative analysis identified themes concerned with crucial aspects of FCNs’ activities, role implementation, and their relationship with patients and families. This study illuminated how the FCN strategically takes care of and identifies patients’ and community needs. Although the COVID-19 outbreak hindered effective FCN project implementation, this study highlighted that the pandemic provided a chance to better identify cultural, organisational and educational weaknesses that need to be addressed to support the full accomplishment of FCNs’ scope of practice.

## 1. Introduction

The COVID-19 pandemic unveiled healthcare systems’ weaknesses, specifically in providing care to frail people. By 2050, the proportion of older adults in Europe will most likely double from 11 to 22% of the total population [1]. In addition, several diseases/conditions are associated with a progressive reduction in functional capacity, leading to a persistent need for care over time. At the European level, these diseases account for roughly 80% of the causes of mortality in people over 65 [2]. Several factors are challenging the delivery of health services; among these are the ageing population and the increase in chronic diseases [3].

This scenario has forced health professionals and organisations to rethink healthcare models and to consider a new paradigm in order to move from a hospital-centred model to one centred on the person and the context in which he or she lives, i.e., the community. It is, therefore, necessary to review how care is delivered in the community. In Italy, the Resolution of the Council of Ministers (Ministerial Decree 71/2022) [4] first and then the Ministerial Decree 77/2022 [5], highlighted the pivotal role of the Family and Community Nurse (FCN) in ensuring nursing care for different levels of health complexity and in collaboration with all the health professionals operating in the same district or area. The COVID-19 pandemic has inevitably highlighted the urgent need to strengthen community services. Critical COVID-19 measures, such as social distancing, lockdowns and travel restrictions, have influenced people’s non-adherence to treatments, generating adverse health outcomes and a lack of care continuity in the community [6]. As a result, global healthcare governments’ initiatives and measures have been increasingly focused on people’s healthcare needs to be managed, planned and delivered in their own homes. Telemedicine systems to support patients in the continuity of care have been primarily implemented to encourage people to stay home. In addition, there is a growing urgency to create and improve models of care tailored to patients, caregivers and families that are economically sustainable and effective [7]. A few studies have been conducted with an ethnographic approach describing the bridging role of the FCN between the general practitioner (GP) and the patient in a community context. The FCN educational and supportive role towards the patient and his or her family context have been highlighted, as well as the FCN strategic role in multi-professional collaborations and prevention initiatives in the community [8,9].

The FCN is the nursing professional involved in the community and continuity of care [10]; thus, FCNs are responsible for nursing in family and community settings and possess specific knowledge and skills in primary care and public health. The FCN acts competently in the provision of complex nursing care, health promotion, prevention and participatory management of individual, family and community health processes, working within the primary healthcare system. FCNs have health as their objective and operate by responding to the health needs of the adult and paediatric populations in a specific territorial and community context and promoting the integration of health and social services [11]. FCNs act with professional autonomy, referring to the nursing services of the district, in close connection with the health services and social services and with the other National Health Service professionals [12].

The FCNs adapt their work according to the different characteristics of the community context and its variety of settings and primary care models. In fact, different settings such as high-intensity urban areas, city suburbs, isolated villages or mountainous areas require specific health organisational and intervention solutions [13]. The FCN profile still represents innovation in the European healthcare national organisations’ context. The European project ENhANCE (European Curriculum for Family and Community Nurses 2018–2021) was founded to create and develop a European curriculum to launch core FCN competencies [14].

FCNs must possess specific competencies, such as establishing new relationships within the care team and citizens, to apply this innovative approach and care model effectively. Nevertheless, implementing new nursing roles in primary care encountered several obstacles, such as resistance to change by medical staff, the absence of legislation recognising the scope of application and an educational system capable of providing nurses with the necessary skills [15]. Although there are several studies on community nursing, the fact of being a role of such a diverse amplitude and one that is embedded in different regulatory, educational and care contexts make it difficult to generalise the experiences of FCNs [16].

### Aim

This study aims to describe the cultural model of FCNs by adopting McGill’s Model of Nursing [17] to assess how the nurse is contextualised in the community setting a few years after the establishment of this new role in Italy. Second, to explore how FCNs and patients perceive the role of FCNs within a primary care implementation project during the COVID-19 pandemic.

## 2. Materials and Methods

### 2.1. Research Design

A focused ethnographic approach was adopted to investigate the beliefs and practices that characterise specific processes of the healthcare setting, practitioners and patients/families [18]; furthermore, it is an approach that focuses on cultures and subcultures framed within a particular context [19,20].

This study was structured following the McGill cultural model [17]. The central components of this model are family, health, learning and collaboration. It directs the nurse to focus on the family and on its strengths and potentials, rather than on its weaknesses or deficits. The model’s goal is to promote the health of the family by creating a supportive learning environment. The overall approach is collaborative and nursing care is adapted to the needs and motivation of the family, taking into account its goals and priorities.

This study was conducted according to the COREQ checklist (see Appendix A) [21].

### 2.2. Setting

The study was carried out in a Community Health Centre (CHC) in a northern Italian region, specifically in the Santhià CHC, within the Local Health authority and territory of Vercelli (Piedmont). Santhià was one of the first NHS districts to implement FCNs’ role. This district is near the researchers’ place of employment; therefore, it was the most appropriate and convenient setting for this project. The Ministry of Health initiative has recently introduced CHCs as a potentially successful primary care model [22]. CHCs are active and dynamic centres for health and well-being for local communities that identify citizens’ needs for healthcare and supply it most appropriately in time and space. The epidemiological and territorial context in which the FCNs in this study operate is a vast geographical area with a high-density resident population. 25% of the resident population is over 65, with an old age rate of 25.65%. The old age rate is a summary indicator of the population’s degree of ageing and the population’s age structure. It is the ratio between the population aged 65 and under 15 per 100 inhabitants. In this area, there are health clusters with a prevalence of chronic diseases and social clusters representing many one-person households [23,24].

The FCNs of this project cared for fragile families and/or singles over 65 living within the Santhia’ territory and registered with the CHC. At this stage of the implementation process, potential patients and families were generally referred to FCNs by the GPs (primarily for therapeutic adherence concerns) and very few by the CHC or other FCNs by word of mouth.

### 2.3. Participant Recruitment

The researchers first contacted the CHC nurse managers and received permission to perform the study, and then FCNs were contacted individually. All FCNs working in the CHC of Santhià participated in the study and they were enrolled once they agreed to sign an informed consent. Some patients were also enrolled to gather their points of view, representations and understanding of FCNs’ work and role in the local healthcare facility. The criteria for the inclusion of patients in the study were as follows: local people, able to communicate in Italian, over 18 years of age and with sufficient cognitive ability to be interviewed. The patients were enrolled and extracted from a database used by the FCNs to take charge of the patients. Once the patients who met the inclusion criteria had been identified, they were contacted by the CHC coordinator, who briefly explained to them the aim of the study and set up an appointment by telephone. Subsequently, the team of researchers met with the patients and informed them about the purpose of the research through an information sheet. Patients signed a consent form to participate in the study and to be interviewed.

During the COVID-19 outbreak, FCNs could not be contacted, and it was impossible to interview them because they were deployed to other healthcare services. Therefore, data collection took place just after the pandemic. On the one hand, FCNs and patients’ memories and reflections captured in the textual material were still very present, while on the other hand, we analysed the qualitative data considering the experiences and difficulties in the project implementation encountered during the COVID-19 outbreak.

### 2.4. Data Collection

Data were collected from June to September 2022 using semi-structured, open-ended interviews. The open-ended questions enable participants to describe their experiences working in patients’ homes and the CHC. Additional questions were used as prompts based on FCN and patient/caregiver responses or as support during the interviews. The interview questions were inspired by the domains of the McGill cultural model [17]; thus, they focused on the FCN in the family setting, the educational role, the identification of needs and health promotion.

Interviews were conducted by EB (Female, PhD, RN Research Fellow) and EP (Male, RN, who previously conducted qualitative studies). The two researchers were methodologically supervised by MB (Female, PhD, RN, Research Fellow) and EDL (Male, PhD, RN, Research Fellow), both lecturers on qualitative research at Italian universities. Interviews were conducted via conferencing software, audio recorded and verbatim transcribed by the interviewer. An interview guide was used (Appendix A). FCNs and patients participated in the interviews in their work, CHC and home environments, in a quiet environment without intrusion from other people. No other people were present during the interviews, except in one interview with the patient in the presence of his son. The researchers observed and described places and the daily activities related to nursing care, following a guide to the taking of ethnographic field notes.

The patient’s interview guide was refined after the first three interviews and started with the following introductory question: Can you tell me about your experience with the FCN concerning the care you need? Other open-ended questions were used to elicit the patients’ views on patterns of care and models of prevention and health promotion oriented towards the population. In a complementary manner, the FCNs’ interview guide explored FCNs’ experience in nurse–patient communication and professional interactions, family and community nursing interventions in managing chronic conditions and health promotion, and challenges experienced by FCNs in their daily practice.

The audio-taped interviews were transcribed with NVivo software. The researchers also checked the completed transcription manually.

During the interviews, EB and EP collected field notes on non-verbal behaviours and nursing documentation for CHC activity management, such as activity calendars, patient care planning and educational interventions [25]. In total, 17 interviews were conducted (5 FCNs and 12 patients).

The expanded observations were also recorded as field notes. After their first field visits, the researchers discussed their observations with the members of the team in order to review the data collection process and generate a contextualised understanding of the cultural pattern. This overcame the problems of observer reliability and bias [26]. The participants also became accustomed to the presence of the researchers, thereby minimising the Hawthorne effect [27]. Data collection continued until data saturation was reached. During the data collection, the research team met through weekly and monthly meetings to allow the field researchers to report on their data collection processes and observations. The observations were collected in a dedicated notebook as field notes, were subsequently reported to the methodological researchers and discussed with the research team members to reach a contextualised understanding of the culture of FCNs in CHC. Finally, a report of the observations was written based on the field notes.

### 2.5. Data Analysis

The focused ethnographic approach requires that data analysis be carried out concurrently with data collection, following a recursive process [19,20,28]. It is essential that “the data are materials to think with” [29]. Based on Spradley (1980), the transcriptions were analysed: domain, taxonomic and component analysis [30]. First, the researchers identified the various possible terms and their semantic relationships by reading and re-reading the interview transcripts, looking for references related to FCNs’ cultural model, and generating open coding independently.

Second, domain analysis worksheets were prepared to structure each domain with the included terms. In the taxonomic analysis, the researchers identified cultural subcategories by searching for similarities among all included terms under the domains. The taxonomic analysis gives “the relationship between all the terms included in the domain”. Then, in the component analysis, attributes assigned to a category were identified and grouped. Researchers achieved categorisation by comparing and contrasting the attributes of each cultural category. Differences in coding were resolved through a comparison session, and codes were consistently compared [31] for their conceptual similarity or attribution of meaning. Regular data session groups were conducted to review the coding [19].

The researchers mutually agreed on the codes and reached a unanimous consensus on how to apply the created codes to the qualitative data [32,33]. The reading, analysis and cyclical comparison work have led to a broader overview that has allowed us to go beyond preconceived beliefs and prejudices in collaborative reflexivity [33,34].

### 2.6. Study Rigour

The research team managed to reach a shared viewpoint on the findings by trying not to allow personal thoughts and previous experiences to interfere during the analysis of the participants’ narratives. Each analysis phase was conducted by at least four researchers (MB, EB, EDL and EB), whose analytical work was collaboratively corroborated in several online meetings.

The inter-analysis team meetings to compare qualitative findings and share sub-themes guaranteed the study’s rigour [35]. As mentioned above, regular data analysis sessions were conducted in which all team members reviewed the coding and reached a consensus on emerging cultural themes. In addition, the authors triangulated data from interviews with patients and FCNs, document reviews and field notes written during observations to better understand the cultural model being studied [19]. Research validity was also supported by a traceable and organised series of cognitive acts that led to theme construction [36]. The researchers all had experience conducting interviews. However, before data collection, the two field researchers still needed to know or interact with the study population.

Finally, a member check [32] was conducted by inviting the FCN participants of this study to attend a meeting. The research team returned the study’s results to FCNs for any insights, possible interpretations and if they felt represented by the emerging themes and issues. The three FCNs who attended the meeting confirmed the accuracy and resonance of the findings.

### 2.7. Ethical Considerations

The study was approved by the Interagency Ethics Committee “A.O. SS. Antonio e Biagio e Cesare Arrigo” of Alessandria (Protocol number 0004936). During the observation, researchers aimed to inform the staff, patients and their caregivers about the study objectives and data collection methods. For the interviews, all participants provided written consent. Participants were informed that they could discontinue the interview at any time and for any reason. The anonymity of the individuals was ensured by avoiding identifiable characteristics in the transcripts.

## 3. Results

A total of 5 FCNs and 12 patients had a face-to-face interview. Most FCNs were female (*n =* 4), and the mean working experience in years as FCNs was 3.5 (range 1.7–4.2). All FCNs’ interviews were conducted in the outpatient nursing clinic and the mean duration was 44 min (range 22–67). The patients interviewed were older people with a mean age of 79.8 years (range 63–95) and one or more comorbidities. Interviews were conducted at the patient’s homes and lasted a mean of 20 min (12–31). The participants’ characteristics are reported in Table 1.

Five main themes were identified: (1) Nurse–patient partnership, (2) Patient empowerment, (3) Caring process, (4) Proactive role, and (5) Barriers to role recognition (Table 2). Illustrative quotes from the interviews and field notes are included below. FCNs and patients have been coded using numbers; ‘FCN’ denotes FCNs and ‘P’ patients.

### 3.1. Nurse–Patient Partnership

This theme highlights the pillars supporting the partnership development between the FCN, the patients and their families and the factors acting as facilitators. The theme has three subthemes: (1) building a trusting relationship over time, (2) a listening attitude and (3) relationship facilitators.

*Building a trusting relationship over time.* The relationship of trust and collaboration between FCNs, patients and their families is essential to establishing a partnership, and it is built over time. The relationship does not exist independently but develops through constant encounters between nurses, patients and their families. This relationship of trust is strengthened over time, making the nurse regarded as a family member.

*It is a relationship that is based on trust since it cannot be otherwise, and it is a relationship that has to be gradually kept alive as if it was a plant always being watered*.(FCN3)

*When I met her* [the FCN] *through my family doctor, when I met her personally, I did not feel that I had met a doctor; I felt like I met a friend. She has cared for me until now, and in all this time, almost a year of knowing her, she has cared for me as a family friend*.(P4)

[Answering the question ‘What does the FCN represent for you?’] *Everything, as if she were my sister*.(P9)

The involvement and participation of the patient in the relationship provides the basis for developing a sense of compassion and trust. Patients’ statements reflect the nurse’s deep knowledge of their health conditions, and the trust established in the relationship is the key to enabling FCNs to enter patients’ homes.

*It’s always the same story, either you have a connection with a person, or you don’t, and with her* [the FCN], *there is much of that, so much so that she knows more about me than I do*.(P1)

*With a relationship made of trust, I can establish an optimal relationship [with the families] and enter their house*.(FCN5)

Although the COVID-19 pandemic imposed the replacement of the FCNs to other community health services in charge of tracking positive cases isolated at home, the relationship previously established with the patients and their families was not compromised by this temporary interruption of contact.

*In October 2021, we* [FCNs] *returned to our pre-COVID role and took care of all the families again. Everyone opened their doors wide to us*.(FCN3)

*A listening attitude.* The FCNs talk and listen to the people being cared for, tailoring their interventions so as to establish a dialogue, but mostly listening to how people live their daily life, their needs and their fears.

*When we* [FCN and patient] *meet, we start talking. I ask the patient what he does when he takes medicines*.(FCN2)

*It’s not just a chat; it’s personal knowledge; you cannot treat a sick person if you don’t know him*.(P1)

The FCNs establish a relationship and adopt active listening; thus, patients share some of the responsibility of caring by expressing their needs. In this process of talking, knowing and trusting, the FCN becomes a reference point. As a result, patients’ sentences reflect a collaborative and supportive relationship in which they can rely on FCNs to contribute to their health and well-being.

*At first, they* [patients and families] *see you only as a nurse, but later, you are seen as a friend with whom to talk, open up and try to solve problems they feel were unsolvable. And then you realise that these are solvable things*.(FCN5)

*He* [FCN] *is my main help because I confide in him. He is a support*.(P3)

*Having a reference person is a great thing, and it’s very helpful*.(P10)

*Relationship facilitators.* Most FCNs stated that ‘being in the community’ facilitates nurse–patient–family relationships. The knowledge of and attendance at community gathering places allows FCNs to involve families and deliver interventions effectively.

*I’m from the neighbourhood, so I’m from the town community where I work. I can know the patients and their families beforehand because I live in the same community*.(FCN2)

[Family involvement] *Can be hindered by distrust or by the fact that sometimes you need to be more settled and grounded in the community. So you do not have enough visibility, you are a ‘stranger’, you are not known, you are an outsider*.(FCN4)

The visibility that FCNs achieve by ‘being in the community’ is an essential factor for the success of interventions at a community level; these are indeed a vector for further awareness of the FCN role and an opportunity to encounter patients and families.

*Once you are known [as an FCN], once you become part of the social structure of a community, then you can make interventions or plan projects because you know where people live [social context]*.(FCN4)

Another facilitating factor for FCN implementation is the role of GPs; they can act as a point of connection and support for building a relationship of trust.

*The GPs were facilitators because they were key to connecting with the families*.(FCN3)

### 3.2. Patient Empowerment

This theme explores how FCNs can help patients gain control over their lives and make decisions. Nurses can play a central role in patient empowerment. The theme consists of two subthemes: (1) sustaining motivation and giving support and (2) asking for feedback and providing counselling.

*Sustaining motivation and giving support.* In the engagement process, FCNs help patients and their families find solutions through a better understanding of their health conditions. FCNs emphasise the importance of patients and their families assuming an active role in achieving the best possible level of health. To be supportive, FCNs require caring attitudes, such as empathy and determination, that can help patients accept the recommendations of healthcare professionals. On the patient’s side, it is very motivating to live their lives beyond their disability or illness.

*We put ‘on the table’ all those elements that can help people to solve their problems and to make their own decisions*.(FCN4)

*Almost always, however, I also have to be firm in letting them know that I am not there as their [Patient] substitute. I cannot substitute them at all*.(FCN5)

[The FCN] *understood my depression and could laugh as I did. She does her job, eh... It’s not that she is here to please me or cuddle me when there is a need she scolds me; I accept it because she does her job, and she does it well... Punctual and precise*.(P1)

*Because it is not that people live to eat and sleep. You also need a small quantity of motivation, and if there is this kind of health professional…You know, there is no-one like her* [FCN] *or almost no one around here*.(P4)

*Asking for feedback and providing counselling.* The FCN promotes patient empowerment through education by providing knowledge and practical skills. In particular, the interviews highlight the role of the FCNs in seeking verbal and practical feedback from patients to assess knowledge transfer. Empowering patients also means that FCNs offer one-to-one feedback to patients and implement ‘reinforcement’ communication to achieve the set goals.

*I ask him [patient] to repeat what I have told him: show me, repeat what I have explained to you as if you had to teach it to me…There is a knowledge component and a gesture component. We ask, for example: How do you measure blood sugar? Where do you prick your finger? What values should you have? How do you take the medications? In addition to repeating information, they* [patients] *must know ‘how to do’ and translate knowledge into action*.(FCN1)

*I ask questions, and from their [patients] feedback, I understand if they have what I have explained*.(FCN2)

*Of course, informative and reinforcing counselling is applied almost every time we meet the patients*.(FCN3)

[Extract from the home visit] *The tone used by the FCN is colloquial and informal. The nurse repeats concepts several times. She removes her mask to make herself more” understandable” and uses gestures to reinforce verbal communication*.(Field notes)

[Extract from the joint visit with the GP in the doctor’s consulting room on Friday morning] *The FCN leads the patient out and summarises the main instructions received from the GP in a quiet place: 1. buy the blood pressure machine and inform the FCN of the purchase in order to schedule a training visit together, 2. go to the pharmacy to have blood pressure taken (Saturday morning) and inform the GP if the values exceed those marked on the medication sheet given by the FCN*.(Field notes)

### 3.3. Caring Process

This theme depicts the process of providing care carried out by FCNs. Emerging aspects are related to the conceptual model adopted and the peculiarities of the nursing process implemented, particularly regarding care setting, information gathering, patient assessment and care delivery modes. The theme comprises three subthemes: (1) family-centred conceptual framework, (2) family’s home is the place for caring and (3) assessing the needs and promoting self-care.

*Family-centred conceptual framework.* A conceptual framework is necessary to guide family and community nursing practices. All FCNs interviewed agree that nursing requires a paradigm shift from the individual-patient approach to a family as a client. The conceptual framework adopted by FCNs is the Calgary Family Assessment Model [37], which considers: (a) the structure of the family, including extended family relationships and outside factors, and (b) its development and functioning, i.e., the family’s ability to perform essential functions. The model also assists the nurse in obtaining information from the family through oral interviews and assessment tools [37].

*Regarding this model* [Calgary], *we have adopted the conceptual framework regarding family assessment*.(FCN4)

*We look at the family and use the Calgary model to carry out the nursing process: there is the whole family to take care of because when one family member is sick, the whole family is also affected*.(FCN5)

The whole family is affected when a family member experiences stress or such an illness. Exposure to COVID-19 infection and the potential transmission between family members living in the same home were not the only worries. The patients also experienced the emotional and physical impacts of isolating family members, especially in the case of different dwellings. The COVID-19 pandemic raised awareness among FCNs about the importance of the family and the connections among its members in managing health problems.

*COVID-19 showed us that the illness of one* [family member] *affects all the others. We learned the importance of looking at things differently, not the single individual, but the whole* [family].(FCN3)

Most of the FCNs adopted the Calgary model. However, some reported that it took much work to understand and adopt it since they did not receive any specific training but were self-taught and had to use English-language reference material.

*At first, it was complicated because none of us is a professional translator, and there were no Italian language textbooks, so we still cannot say that we are experts in the Calgary model*.(FCN4).

*Family’s home is a place for caring.* The home visits represent a shift in the nurse–patient relationship from what occurs in the hospital setting because the nurse is a guest in the patient’s home. Through home visits, FCNs can observe the physical setting and get a picture of family life. The home environment ‘speaks’ about who lives there and, with specific regard to FCNs’ perspective, it can affect patients and their family’s health.

*I really like to go to the patients’ homes because that is where their life is. I can discover their stories inside each family’s home by looking at the photographs displayed. I can understand much more; I can gather more information than in being at the FCN outpatient clinic*.(FCN3)

*Assessing the needs and promoting self-care*. Within the FCNs’ caring processes, the GP functions as a gateway. Indeed, the GP is the first to report the case, sending an email to the FCN and providing initial information about the patient. Then the FCNs meet the patients, ask about their daily habits, and use assessment tools to evaluate functional status, falling risk, medication adherence and lifestyle.

*If the GP has reported the case, he sends us a formal referral on the institutional email. Once we have read the referral, we call back the general practitioner to gather further information*.(FCN1)

*The information we ask for is mostly lifestyle information in general (...) related to medication and nutritional aspects, physical exercise, and lifestyle*.(FCN2)

*We assess* [the patient] *using different tools: the sunfrail tool, the ADL and IADL scale, the Morinsky scale and so on...I assess the person and then the family*.(FCN5)

The FCNs also perform assessments to understand the family’s overall situation, both in terms of its functioning and heritable traits. FCNs use as much information as possible to obtain a complete picture of patients and their family context, including available health data.

*Then we do a family assessment to understand its structure, development, and internal relations*.(FCN1)

[We use] *a family history form and genogram to learn about the family links, the diseases each family member has and their heritability*.(FCN5)

*We have access to the hospital databases with all the patients’ data. I look for information regarding disabilities that can help me to understand who I will be dealing with. I also search for information about the composition of the family*.(FCN1)

FCNs have to deal with clinical situations characterised by chronicity, teach patients how to self-care to adopt healthy lifestyles and safe behaviours, and provide information and support material (example from the field notes: *medication sheet, dietary instructions*). Furthermore, considering the patient’s advanced age and weak attitude towards change, FCNs seek to propose gradual and sustainable directions for improving their lifestyles. From the patient’s perspective, this means understanding that the guidance they are given is effective and can make a difference in their quality of life.

*We teach them* [patients] *to self-manage by adopting correct lifestyle behaviours: eating healthy food, making the home environment safe, using closed shoes, and removing carpets*.(FCN3)

[We teach] *a few specific things that they* [patients] *can start doing; we don’t list things to do or not to do. For example, we ask them to keep a food diary, writing down for 2–3 days what they eat, how they cook, how much they drink and whether they are hungry*.(FCN2)

*As soon as he* [FCN] arrives, he asks about my situation…Are you fine or not? Have you eaten or not? Did you drink? Did you take your medicines? They [FCNs] *teach many things that make one feel glad. You can know many things but not the ones you may need at that moment*.(P5)

### 3.4. Proactive Role

This theme highlights the proactive role of FCNs in nursing care and the development of collaborative networks. This theme consists of two subthemes: (1) promoting prevention and healthy behaviours at the family and community levels and (2) pursuing interprofessional collaboration.

*Promoting prevention and healthy behaviours—family and community level.* Proactive nursing involves the early detection of health problems at the family and community levels. At the family level, the FCN provides anticipatory guidance by providing appropriate information about healthy lifestyles and disease prevention.

*This perspective of anticipatory guidance puts us in a trusting relationship with the family. Thus, it allows us to identify key risks preliminarily. You can guide the family in the process of becoming aware*.(FCN4)

The proactive role is maximised when the FCN meets the community. During health promotion events, FCNs implemented a range of activities aimed at introducing themselves, providing information, orienting citizens to the services available at the CHC and advising about the professionals at their disposal, such as medical specialists, home care nurses, social care workers or psychologists. However, the FCNs stated that there needed to be more opportunities to implement preventive measures through nursing programmes at the community level. A significant obstacle to community interventions is related to organisational structures: a lack of resources, including time and adequate spaces.

*We intend to organise educational events in our town. We could intervene to give information on lifestyles*.(FCN2)

*We would love to do these open days to address health issues such as diabetes, smoking, and alcoholism. However, we would need suitable spaces and places to have someone to help us with advertising*.(FCN5)

The difficulty in carrying out health promotion programmes, even at the community level, highlights a limitation in adopting the Calgary model [37], which aims to consider the reality of the person, family and society in an interrelated way. The interventions led by our FCNs are directed to empower mainly the patients, secondarily the family members and, only later on, the community.

*Pursuing interprofessional collaboration.* The FCN proactive approach is also realised in the field of interprofessional relationships. FCNs meet with other professionals daily and look for strategic community stakeholders. Building functional relationships to achieve the health goals of patients and families is part of their practice. From the researchers’ field notes and the FCNs’ own words, it emerges that the organisation and the physical location in which they are placed may favour collaboration with other professionals.

*We collaborated with many stakeholders, starting with the GPs, social workers, psychologists, home care colleagues, and the administrative staff of the various services. The FCNs interact with mayors, voluntary associations, the third sector and the priest of the municipality*.(FCN1)

*The outpatient nursing clinic within the CHC facilitates because we work close to the GPs of the families we care for and the specialists, the social services, and the home care colleagues with whom there are frequent exchanges of information*.(FCN3)

*The social worker shares the office with the FCNs and has her desk and computer workstation. Sharing the space facilitates close collaboration between the two professionals, and the exchange of information on the patients follows jointly*.(Field notes)

Most FCNs recognise the added value of proximity to other healthcare professionals, as it promotes in-person communication. Healthcare professionals who share the same care goals are likely to create an informal relationship in the first instance because it facilitates dialogue and trust building. The FCNs stated that informal relationships were easier to establish and initially facilitated their inclusion in the organisational structure. As their role became more settled, FCNs began to feel the need to strengthen these relationships, formalising them into structured networks that could ensure a shared vision of patient care and lay the foundation for building a shared care pathway.

*Currently, our networks are informal. Some networks were also built on a ‘gentleman’s word’. We are trying to build stronger networks that can become defined and structured*.(FCN1)

*Informal relationships facilitated us so much more than formal relationships*.(FCN4)

During the COVID-19 pandemic, FCNs were deployed to other community services, focusing on monitoring positive cases at home. At the same time, they tried to keep connected with their patients, primarily by making phone calls, asking them how they were doing and solving their problems at a distance and tailoring care to the needs of the pandemic. A highly collaborative working climate facilitated the demand for fast adaptation to new tasks and caring modes experienced by FCNs. The relationships established among different professionals, services and institutions made it possible to solve problems of a very diverse nature rapidly. In fact, it was possible to provide isolated citizens with medicines and food, activate psychological support and even restore electricity to those left without supply, thanks to the strong will of all stakeholders to work towards a common goal.

These powerful collaborative dynamics overcame the usual bureaucratic procedures and consolidated hierarchies often linked to the role of GPs. Indeed, the pandemic has challenged the physician-centric view, which focuses on curing the individual’s illness rather than caring for the entire family unit in managing each member’s health issues.

*I really appreciated the COVID-19 time. I liked it so much because everything was possible. So, just with a phone call, you establish relationships and collaborations with the ‘rest of the world’*.(FCN3)

*It was much easier in the pandemic context because all hierarchies were cancelled. There was an effective collaboration to ‘responding’ [to care needs] because the responsibilities and intentions were going in the same direction. That certainly was important for us*.(FCN4)

As of October 2021, FCNs returned to their pre-pandemic position, finding that many households had changed. Some older adults died or were institutionalised, and many families became single-person households. The FCNs feared it would be difficult to recover their relationships with their patients, but this was not the case. The real problem was the widespread change in the household structure that required FCNs to deal with the lack of a family environment, preventing the complete application of the conceptual framework and, at the same time, reinforcing awareness of the central role of the family.

### 3.5. Barriers to Role Recognition

In this theme, the FCN’s role recognition is linked to organisational factors and staff and patient awareness. Major emerging sub-themes regarding the barriers are (1) lack of understanding/awareness of the FCN role and (2) low organisational readiness for change.

*Lack of understanding/awareness of FCN role.* FCNs reported that they had to cope with a challenging initial situation. They perceived that they needed more managerial support at the beginning of their assignment, including formal recognition within the organisation. FCNs felt invisible in the eyes of service management and poorly understood by their nursing colleagues who did not fully grasp the FCNs’ scope of practice. FCNs complained about the lack of awareness of their role among nursing colleagues, other health professionals and patients, who only knew FCNs when the GPs introduced them. In addition, the physical environment of the CHC did not ensure the proper visibility of the FCN service.

*The local health system does not recognise and appreciate us*.(FCN1)

*The FCN is not known either by the nurses themselves or, even worse, by other health care professionals. So, ignorance about who the FCN is does not facilitate the relationship* [with other professionals].(FCN2)

*The doctor sent me upstairs to your office. I did not even know you* [FCN] *were there; I did not know*.(P3)

[Atrium of the CHC] *the information placards need indications about the location of the FCN outpatient clinic and the route to it*.(Field notes)

FCNs reported that it is difficult to make visible what they do, as their technical skills are not the predominant ones in their daily activities. Consequently, patients and professionals need help understanding the FCN role.

*In addition, it is not easy to explain it when they ask you because the FCN has a breadth of scope of practice that is not understood by those who work only on curing and think exclusively of techniques or actions. So it is not easy to make people understand*.(FCN2)

*The patients themselves initially did not understand what we were doing because we only sometimes go to their homes to measure blood pressure, give an injection or take a blood sample. Not being able to quantify what we do, is the hardest part. To demonstrate the work behind every family cared for without having performed technical interventions is a challenge*.(FCN5)

*Low organisational readiness to change.* The healthcare system’s organisational culture, which still focuses on disease treatment in acute care settings, does not facilitate the change needed to enable FCNs to reach the full potential of their roles. As a result, FCN‘s role implies a paradigm shift in reorienting health services to the person rather than the disease. This shift will entail the involvement of individuals, families and the community in the overall healthcare process.

*We have to implement a different service. Otherwise, we are replicating what is done in the hospital for people who are sick...If a person is healthy or has chronic issues under control, they do not need the same care provided in a hospital...Healthy people should be helped to maintain their health*.(FCN4)

Although the pandemic challenged the physician- and hospital-centric view of healthcare, some FCNs reported that their practice is still overshadowed by the dominant bio-medical model, which prioritises curing the disease rather than empowering the person. In contrast, FCN’s role is to promote the adoption of healthy lifestyles and thus maintain patients’ well-being.

*None of the settings we work in* [CHC] *has a family-centred orientation, which is a first problematic aspect, because we act inside a treatment facility, and the treatment is almost always individual*.(FCN4)

## 4. Discussion

This study explored the FCN cultural model and how this role has been implemented, especially during the COVID-19 pandemic. Ethnographic lenses have helped to understand the cultural model of family and community nursing in a small area in northern Italy covered by the NHS. The four overarching themes of our analysis were critically presented as the intertwined voices of patients and FCNs in this innovative service. Patients’ and FCNs’ themes overlapped, showing a convergence of how the novelty was perceived and what elements characterise the cultural context of the family and community nursing in that area. As mentioned already, the FCN’s scope of practice is to address the health needs of a specific community’s adult and paediatric population by promoting the health and social integration of services [11,38,39].

Historically, nursing services at the local level were provided by district nurses. In contrast, the FCN represents an evolution of this concept, and NHS initiatives support it and have been highly valued since the pandemic [4,5,10,40].

The first overarching theme is about the ‘nurse–patient partnership’. The FCNs of this study approached the individuals and their relatives becoming, at some point, *part of the family,* showing that the axiom of community and family interventions was accomplished. FCNs showed a willingness to embrace their role and to strive to implement the concept of PHC from a holistic and relational point of view. The presence of the FCN who entered the patients’ homes providing support, motivation and health education gave space for new meanings. It facilitated the caring relationship, as well as its values. Building trust between FCNs, patients and their families at home and in primary care settings is more fundamental than ever in community nursing. Several studies in the field show how trust improves therapeutic adherence and continuity of care in the community setting [41] and how the nursing figure could replace the GP in some respects [42].

The second central theme regards patient empowerment. FCNs in the present study focus on promoting self-care, supporting patients’ and families’ autonomy, enhancing their understanding and assisting them in acquiring the skills to manage their health issues better, as other studies point out [43].

The strength of FCNs’ role, especially at the beginning and during its implementation, was to propose a less instrumental and procedural approach to care, typical of district nursing, while supporting the population with different nursing and communication skills [44].

From this, FCNs were able to develop and support patient empowerment with small but continuous and effective interventions while adopting an active listening and troubleshooting attitude. According to other studies [45], communication skills have emerged as a critical characteristic of FCNs that can make a difference in patient, family and community engagement [46,47]. FCNs must communicate with and listen to their patients without passing judgement and must work hard to provide objective, individualised treatment with the utmost compassion and empathy [48]. Meaningful experiences have been undertaken at the European level in patient-centred care [49,50] and in Italy recently. The first guideline was adopted concerning the modalities of participation in the decision-making processes of the Ministry of Health by citizens’ and patients’ associations or organisations involved in health issues [51].

Regarding the ‘caring process’ theme, the FCNs interviewed declared having adopted the family-centred Calgary model [37]. This model relates perfectly with the principles of the McGill cultural model [17], which focuses on the family as the setting where the FCN provides care through multi-professional collaboration. Although the FCNs agree on the effectiveness and comprehensiveness of the Calgary model, some needed help translating it from theory to practice, admitting an initial struggle in dealing with a foreign model and the need to be trained by experts. Other studies have reported adopting different FCN organisational models [52,53,54], all featuring the aim of providing patient–family–community-centred care while highlighting the need for future research on the implementation of family and community nursing care models [10].

In the present study, the population cared for by the FCNs was aged predominantly, single and frail. In such a context, FCNs help individuals cope with illness and chronic disabilities by advising on lifestyle and behavioural risk factors. The involvement of FCNs in health promotion, including primary, secondary and tertiary prevention, has been widely reported in the recent literature [10]. Concerning the prevention of long-term disabilities, FCNs support patients, families and communities in disease management, such as vascular and neurological issues [55,56], diabetes [57], chronic heart failure [58] and chronic pain management [59,60]. Through prompt detection, the FCN can ensure that the health problems of older citizens with long-term and chronic conditions are treated early, fostering complication management and hospital avoidance [61].

From the subtheme ‘promoting prevention and healthy behaviours—family and community level’, within the ‘proactive role’ theme, emerges another limit in adopting the Calgary model, i.e., the difficulty to go beyond the patient perspective and also reach the family and, especially, the whole community to the same extent. A possible explanation for such difficulties might be related to the characteristics of the population. In the present study, FCNs were asked to care for many patients living alone, and their interventions mostly regarded an individual empowering process rather than a family or societal one. Most of the patients interviewed lived independently with relatives or social services support. The pandemic exacerbated this condition, which further increased single-person households [10].

An interesting aspect that emerged concerns the type of interprofessional collaboration characterising the implementation of family and community nursing in the Vercelli area at the time of the COVID-19 outbreak. Building functional relationships to achieve the health goals of patients and families is part of FCN practice. However, the demand for fast adaptation to new tasks and effective responses imposed by the pandemic strengthened interprofessional collaboration, overcoming established hierarchies and fostering the shift away from the physician-centric view of care [40,62,63]. The pandemic challenged the FCN organisation, but at the same time, it represented an opportunity to experience how much easier organisational changes can be when all professionals involved have the same goal and this prevails over hierarchy, as reported in other healthcare settings [64].

The last theme is about barriers to role recognition. FCNs complained about a lack of awareness of their role among nursing colleagues and other health professionals and felt invisible at the beginning of their mandate. Although the scope of practice of FCNs has been clearly outlined and reinforced by governmental [4,5] and local initiatives, and a position paper [11] was shared between the Italian Nursing Council and the Ministry of Health, there are still many obstacles to implementing FC nursing. The FCNs also complained about low organisational readiness for change. The NHS organisational culture, historically focused on disease treatment in acute care settings, does not facilitate the changes needed to enable FCNs to reach the full potential of their roles [65]. In this regard, the pandemic, increasing single-person households and thus people living alone, emphasised the need to develop primary care services and the role played by the FCN. According to Poghosyan [66], COVID-19 highlighted the potential benefits of full practice authority for advanced nursing roles in primary care, as it improves access and quality of care and patient outcomes.

Overall, the obstacles to model implementation that emerged in the present study are aligned with the current literature, always concerned with the struggles encountered in creating and promoting new roles and professional figures in the context of PHC [67,68,69,70]. As other studies reported, the cultural change to improve the organisational development of primary care should be supported by proper education to the FCN role, comprising specific communication skills and competencies in project management, epidemiology, disease prevention and health promotion at the community level; furthermore, by the possibility of introducing this core knowledge into postgraduate specialisation courses and even bachelor’s degree curricula [70,71,72,73,74].

### Strengths, Limitations and Future Challenges

This study describes the FC nursing cultural model in a specific Italian context in northern Italy, where the organisational and regulatory background is still in a process of change. Moreover, the patient interviews were conducted with aged patients with comorbidities who gave their consent, depicting the experience of a frail elderly group. Thus, both the study’s context and the qualitative approach do not allow generalisations. However, with regard to transferability, the in-depth description provided in the results section can provide stakeholders and researchers with an interesting contribution to understanding similar experiences.

An additional limitation might be that the research team mainly comprises nurses with a PhD and experience in qualitative research. However, an attempt was made to maintain the methodological rigour of the study design as much as possible.

In terms of future challenges, the present study shows that further research is needed to determine FCNs’ health issue management, including the type of context in which they operate and the conceptual model they adopt. Studies directed at understanding the specific epidemiological and psycho-social aspects of the population cared for by the FCNs will help to raise awareness of the population’s needs. As a result, this will help to adequately address FCNs’ collaboration with other professional figures, which can be strategic in implementing appropriate holistic care.

Finally, future challenges also involve FCN education. FCNs should possess valid tools for effective health promotion in the community through appropriate training and by adopting an active and reflective process on the care model and local organisational aspects.

## 5. Conclusions

This is the first Italian study on this topic that adopts a focused ethnographic methodology. The present study depicts how the FCN strategically takes care of and identifies patients and community needs in the Santhià CHC, within the Local Health authority and territory of Vercelli (Piedmont). The results highlight that although the activation of the FCN started some years ago, there are still difficulties in implementing this new role. Patients describe FCNs as a reference figure for their care pathway, but the recognition of FCNs’ potential at the organisational and practical levels still needs to be improved. FCN role implementation does not only depend on the competencies put in place by the health professionals, but it is also related to the specific epidemiological context and the local health organisational structure and culture. The COVID-19 outbreak was an additional factor that hindered effective FCN project implementation, but it provided an opportunity to better identify cultural, organisational and educational weaknesses that need to be addressed to support the full accomplishment of FCNs’ scope of practice.

## Figures and Tables

**Table 1 ijerph-20-01948-t001:** Participants’ characteristics.

Family and Community Nurses (*n =* 5)	
Female gender (N, %)	4 (80)
Age, years (mean, range)	48.2 (43–56)
Overall working experience, years (mean, range)	22.9 (13.6–34.1)
Working experience as FCN, years (mean, range)	3.5 (1.7–4.2)
**Patients and caregivers (*n =* 12)**	
Female gender (N, %)	7 (58.3)
Age, years (mean, range)	79.8 (63–95)
Marital status	
Married (N, %)	1 (8.3)
Divorced (N, %)	1 (8.3)
Widowed (N, %)	9 (75)
Single (N, %)	1 (8.3)
Numbers of comorbidities	
1 (N, %)	1 (8.3)
2 (N, %)	1 (8.3)
3 (N, %)	3 (25)
4 (N, %)	5 (41.7)
6 (N, %)	2 (16.7)
Frequent comorbid conditions	
Hypertension (N, %)	8 (66.7)
Diabetes (N, %)	5 (41.7)
Osteoporosis (N, %)	4 (33.3)
Heart disease (N, %)	4 (33.3)
Chronic obstructive pulmonary disease (N, %)	2 (16.7)
Hypothyroidism (N, %)	2 (16.7)
Hypercholesterolemia (N, %)	2 (16.7)

**Table 2 ijerph-20-01948-t002:** Themes and subthemes emerging from interviews and field notes.

Themes	Subthemes	Perspective
Nurse–patient partnership	Building a trusting relationship over time	FCN + P
	A listening attitude	FCN + P
	Relationship facilitators	FCN
Patient empowerment	Sustaining motivation and giving support	FCN + P
	Asking for feedback and proving counselling	FCN + RFN
Caring process	Family—centred conceptual framework	FCN
	Family’s home is the place for caring	FCN
	Assessing the needs and promoting self-care	FCN + P + RFN
Proactive role	Promoting prevention and healthy behaviours—family and community level	FCN
	Pursuing interprofessional collaboration	FCN + RFN
Barriers to role recognition	Lack of understanding/awareness of FCN role	FCN + P + RFN
	Low organisational readiness to change	FCN

Legend: FCN—Family and Community Nurse, P—Patient, RFN—Researchers’ Field Notes.

## Data Availability

The data are not publicly available due to restrictions (e.g., privacy and limitations in consent about sharing data with others).

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
