# Peer review of "The Family and Community Nurses Cultural Model in the Times of the COVID Outbreak: A Focused Ethnographic Study"

_ijerph, 2023, doi:10.3390/ijerph20031948_

Round 1

Reviewer 1 Report

Please see review in word doc attached

Author Response

Dear reviewer,

We appreciate your careful review and the suggestions you gave us to improve this manuscript. We have considered each comment and made all changes required, including English proofreading.

We include with our revised manuscript an itemized, point-by-point response to your precious comments.

Reviewer 2 Report

The study presented by the authors is based on the application of only 17 interviews, although the appropriate protocols were considered, the number would seem to be low and it would be necessary to check that it is representative; however, the findings they report are very general, which from my point of view would only arouse a medium interest in the scientific community. In the same way, I propose to make the following adjustments to your document:

It is necessary to add the literature review section and describe the most important topics of your study according to what has been reported.

Rename the Aim subsection as Research gap and add information that allows a clearer and more summarized view of the information gap that the study intends to address. Here you can also add the objectives, as well as some research questions if considered appropriate.

The information that the authors are presenting in the discussion section is more related to literature review and conclusions of their results; however, they are not discussing their results comparing them with other similar studies that have been reported, similarities and differences. This section should be rewritten.

In the limitations section, it would be appropriate to add future lines of research.

Author Response

(The authors gave the same response as above.)

Round 2

Reviewer 2 Report

The authors have addressed my comments appropriately. I consider that the work is now better written, so I recommend its publication.